# Integrin β₃ directly inhibits the Gα₁₃-p115RhoGEF interaction to regulate G protein signaling and platelet exocytosis

Yaping Zhang [1,4], Xiaojuan Zhao[1,4], Bo Shen [1,4], Yanyan Bai [1], Claire Chang[1], Aleksandra Stojanovic[1,2], Can Wang [1], Andrew Mack [1], Gary Deng[3], Randal A. Skidgel[2], Ni Cheng [1] & Xiaoping Du [1] ✉

The integrins and G protein-coupled receptors are both fundamental in cell biology. The cross talk between these two, however, is unclear. Here we show that β₃ integrins negatively regulate G protein-coupled signaling by directly inhibiting the Gα₁₃-p115RhoGEF interaction. Furthermore, whereas β₃ deficiency or integrin antagonists inhibit integrin-dependent platelet aggregation and exocytosis (granule secretion), they enhance G protein-coupled RhoA activation and integrin-independent secretion. In contrast, a β₃-derived Gα₁₃-binding peptide or Gα₁₃ knockout inhibits G protein-coupled RhoA activation and both integrin-independent and dependent platelet secretion without affecting primary platelet aggregation. In a mouse model of myocardial ischemia/reperfusion injury in vivo, the β₃-derived Gα₁₃-binding peptide inhibits platelet secretion of granule constituents, which exacerbates inflammation and ischemia/reperfusion injury. These data establish crucial integrin-G protein crosstalk, providing a rationale for therapeutic approaches that inhibit exocytosis in platelets and possibly other cells without adverse effects associated with loss of cell adhesion.

G protein-coupled receptors (GPCRs), a family of cell surface receptors with seven-transmembrane domains, detect molecular stimuli outside of the cell and activate intracellular responses[1]. GPCRs comprise the largest known gene family and are frequently targeted for drug development[2,3]. They transmit signals by coupling with intracellular heterotrimeric G proteins and upon receptor ligation, GPCRs function as guanine nucleotide exchange factors (GEFs), that convert the α subunit (e.g. Gα_s, Gα_{i/o}, Gα_{q/11} and Gα_{12/13}) of a Gαβγ protein complex from the GDP-bound inactive form to the active GTP-bound form. Activated Gα subunits dissociate from the β/γ subunits, and both interact with their respective downstream targets to transmit GPCR signals[4,5].

GPCR signaling stimulates and regulates cell adhesion, spreading, contractility, migration, phagocytosis, and exocytosis, in which the integrin family of α:β heterodimeric adhesion receptors play a crucial role[6–8]. Certain GPCRs, such as receptors for thrombin and thromboxane A2 (TXA2) stimulate Gα₁₂/Gα₁₃ binding to the RGS domain of Rho guanine nucleotide exchange factors (RhoGEF), such as p115RhoGEF. This activates the RhoGEF to convert inactive GDP-bound small GTPase RhoA into the active GTP-bound form[9,10], and thus stimulates contractility, assembly of actin stress fibers[11] and promotes exocytosis in platelets[12,13] and other cells types[14,15]. However, Gα₁₃ is dispensable for the activation of integrins[16,17]. GPCRs, via Gα_q and Gα_i signaling pathways, stimulate "inside-out" signaling, leading to the binding of talin and kindlins to the cytoplasmic domain of integrin β subunits and consequent activation of the ligand binding function of integrins, which mediate cell-matrix and cell-cell adhesion[18–20]. Ligand binding to integrins conversely induces "outside-in" signaling leading

[1]Department of Pharmacology and Regenerative Medicine, University of Illinois at Chicago, Chicago, IL 60612, USA. [2]Dupage Medical Technology, Inc., Chicago, IL 60612, USA. [3]Eli Lilly, Indianapolis, IN 46285, USA. [4]These authors contributed equally: Yaping Zhang, Xiaojuan Zhao, Bo Shen. ✉e-mail: xdu@uic.edu

to cell spreading, retraction, migration, and exocytosis[8,21] (Supplementary Fig. 1). Despite the importance of talin in inside-out signaling and in the late retraction phase of outside-in signaling, the early phase of integrin outside-in signaling leading to cell spreading does not require talin binding to integrins[22,23], but requires $G\alpha_{13}$, which interacts with the ExE motif conserved in the cytoplasmic domain of several integrin β subunits including $\beta_1$[24], $\beta_2$[25] and $\beta_3$[17,23]. This interaction is critical for the integrin-dependent activation of c-Src and c-Src-dependent downstream outside-in signaling pathways[17,26]. Thus, G proteins regulate the adhesion and signaling functions of integrins. However, it is unclear whether integrins or integrin signaling regulates GPCR-mediated intracellular signaling. In particular, it is unknown whether the direct interaction between integrins and $G\alpha_{13}$ can regulate $G\alpha_{13}$-coupled GPCR signaling.

In blood platelets, GPCR and integrin signaling pathways are both important in stimulating granule secretion of prothrombotic, proinflammatory, and pro-proliferation factors/receptors[27]. These factors and receptors are crucially involved in occlusive thrombosis and in facilitating inflammation, atherosclerosis[28,29], and ischemia/reperfusion injury[30]. GPCRs stimulate integrin-independent granule secretion prior to integrin activation[31]. Following integrin ligation and primary aggregation, integrin outside-in signaling induces integrin-dependent granule secretion[20]. Whereas $G\alpha_{13}$ mediates both GPCR-dependent RhoA signaling and integrin outside-in signaling, the relationship between $G\alpha_{13}$ and the complex integrin-dependent and integrin-independent pathways of granule secretion is not clear. Understanding the role of $G\alpha_{13}$ and the regulatory mechanisms of these pathways may facilitate innovative therapeutic approaches to selectively modulate platelet granule secretion independently of platelet adhesion/aggregation and hemostatic function, which current antiplatelet drugs fail to do.

In this work, we demonstrate that binding of the integrin $\beta_3$ cytoplasmic domain to $G\alpha_{13}$ through its ExE motif, directly disrupts the interaction between p115RhoGEF and $G\alpha_{13}$, thus inhibiting $G\alpha_{13}$-p115RhoGEF-mediated RhoA activation and signaling. In platelets, this inhibitory effect of integrins greatly reduces GPCR-mediated integrin-independent granule secretion. On the other hand, the $G\alpha_{13}$-integrin interaction stimulates integrin outside-in signaling and integrin-dependent granule secretion. Furthermore, we demonstrate that the inhibitor peptide derived from the $G\alpha_{13}$-binding $\beta_3$ ExE motif disrupts the $\beta_3$-$G\alpha_{13}$ interaction to inhibit both GPCR-induced integrin-independent and integrin outside-in signaling-dependent granule secretion. This dual effect results in robust inhibition of granule secretion without affecting primary platelet aggregation, suggesting a new therapeutic approach for the development of selective inhibitors of platelet granule secretion for treating thrombo-inflammatory diseases.

## Results

### Direct inhibition of $G\alpha_{13}$ binding to the RGS domain of p115RhoGEF by the integrin $\beta_3$ cytoplasmic domain

To explore a possible relationship between integrin-$G\alpha_{13}$ interaction and p115RhoGEF-$G\alpha_{13}$ interaction, we developed an in vitro binding assay in which the purified recombinant RGS domain of p115RhoGEF was allowed to bind to purified recombinant $G\alpha_{13}$ in the absence or presence of increasing concentrations of purified recombinant $\beta_3$ cytoplasmic domain ($\beta_3$ CD)-GST fusion protein. $\beta_3$ CD dose-dependently inhibited binding of the RGS domain of p115-RhoGEF to $G\alpha_{13}$, whereas the control GST protein had no inhibitory effect (Fig. 1a). To determine whether this inhibition was due to the known $\beta_3$ ExE motif interaction with $G\alpha_{13}$[23], we used a myristoylated synthetic peptide with a sequence identical to the $\beta_3$ ExE motif, mP6 (Myr-FEEERA), and found that it dose-dependently inhibited the binding of the p115RhoGEF RGS domain to immobilized $G\alpha_{13}$ (Fig. 1b). It also inhibited the binding of $G\alpha_{13}$ to the immobilized p115 RGS domain as compared

with the control peptide (Fig. 1c). These data demonstrate that the ExE motif in the cytoplasmic domain of $\beta_3$ directly inhibits the binding of $G\alpha_{13}$ to p115RhoGEF in in vitro assays using purified proteins.

### The inhibitory role of integrin $\beta_3$ on GPCR-stimulated $G\alpha_{13}$-p115RhoGEF interaction in platelets

To investigate whether the $\beta_3$ inhibition of $G\alpha_{13}$-p115RhoGEF interaction occurs in a cellular system, we compared the co-immunoprecipitation of $G\alpha_{13}$ with p115RhoGEF in wild type (C57BL/6J) platelets and in integrin $\beta_3^{-/-}$ platelets with or without stimulation of platelets with GPCR agonist thrombin. Thrombin is known to activate the $G\alpha_{13}$-p115RhoGEF-RhoA signaling pathway[10]. To avoid possible effects of secondary $TXA_2$ receptor-mediated $G\alpha_{13}$ activation, platelets were pre-treated with aspirin to block $TXA_2$ synthesis[32]. Only a low level of $G\alpha_{13}$ was co-immunoprecipitated with p115RhoGEF in wild-type unstimulated platelets as compared with IgG control (Fig. 1d,e). Within 1 min following thrombin stimulation, $G\alpha_{13}$ co-immunoprecipitation with p115RhoGEF was enhanced in wild-type platelets (Fig. 1d, e). Remarkably, co-immunoprecipitation of $G\alpha_{13}$ with p115RhoGEF was almost 3-fold higher in $\beta_3^{-/-}$ platelets as compared with wild-type platelets following thrombin-stimulated platelet activation (Fig. 1d, e). Of note, the total expression of $G\alpha_{13}$ or p115-RhoGEF was not changed in $\beta_3^{-/-}$ platelets as compared with wild-type platelets (Supplementary Fig. 2). Clearly, $\beta_3$ expressed in wild-type platelets had a significant inhibitory effect on the $G\alpha_{13}$-p115RhoGEF interaction, which was abolished in $\beta_3$ knockout platelets. To further determine if and how ligand binding to integrin $\alpha_{IIb}\beta_3$ would affect $G\alpha_{13}$-p115RhoGEF binding, we tested the effect of integrin antagonist RGDS peptide on the co-immunoprecipitation of $G\alpha_{13}$ with p115RhoGEF, using RGES peptide as a control (Fig. 1f, g). Similar to $\beta_3$ knockout, RGDS-treatment enhanced $G\alpha_{13}$-p115RhoGEF co-immunoprecipitation following platelet activation, relative to control RGES-treated platelets (Fig. 1f, g). In control RGES-treated platelets, $G\alpha_{13}$-p115RhoGEF interaction was reduced following the initial wave of binding (Fig. 1f, g). In contrast, $G\alpha_{13}$-p115RhoGEF binding in RGDS-treated platelets was persistently high after the initial increase (Fig.1f, g). Similarly to RGDS, the clinically used integrin antagonist Eptifibatide (Integrilin) also enhanced thrombin-induced $G\alpha_{13}$ binding to p115RhoGEF in platelets (Supplementary Fig. 3). These data suggest that blocking the ligand binding to integrin $\alpha_{IIb}\beta_3$ likely also diminished the ability of $\beta_3$ to inhibit $G\alpha_{13}$-p115RhoGEF interaction. Thus, the function of $\beta_3$ in inhibiting $G\alpha_{13}$-p115RhoGEF interaction is stimulated by ligand binding to integrin $\alpha_{IIb}\beta_3$.

We next determined whether the ExE peptide mP6 would inhibit the $G\alpha_{13}$-p115RhoGEF interaction in thrombin-stimulated platelets similarly to the purified assay system. When mP6 was pre-incubated with wild-type mouse platelets followed by thrombin stimulation, co-immunoprecipitation of $G\alpha_{13}$ with p115-RhoGEF was almost totally abolished (Fig. 1h, i). Thus, the ExE motif in the $\beta_3$ cytoplasmic domain inhibits $G\alpha_{13}$-p115RhoGEF interaction during thrombin-stimulated platelet activation.

### The role of integrin $\beta_3$ in inhibiting thrombin-induced RhoA activation

A direct downstream consequence of $G\alpha_{13}$-p115 RhoGEF interaction is the activation of RhoA[10,33,34]. We thus investigated whether integrin $\beta_3$ regulates $G\alpha_{13}$/p115RhoGEF-mediated RhoA activation during GPCR-induced platelet activation using a GTP-bound RhoA pull down assay. As expected, thrombin-stimulated RhoA activation in wild-type platelets, which was transient, peaked after 1–2 min and then declined to near background level (Fig. 2a, b). Thrombin-stimulated RhoA activation was enhanced in $\beta_3^{-/-}$ platelets compared to wild-type platelets (Fig. 2a, b), consistent with the enhanced $G\alpha_{13}$-p115RhoGEF interaction shown in Fig. 1d. Importantly, $\beta_3^{-/-}$ platelets stimulated with thrombin displayed prolonged RhoA activation persistent throughout the time course instead of the transient RhoA

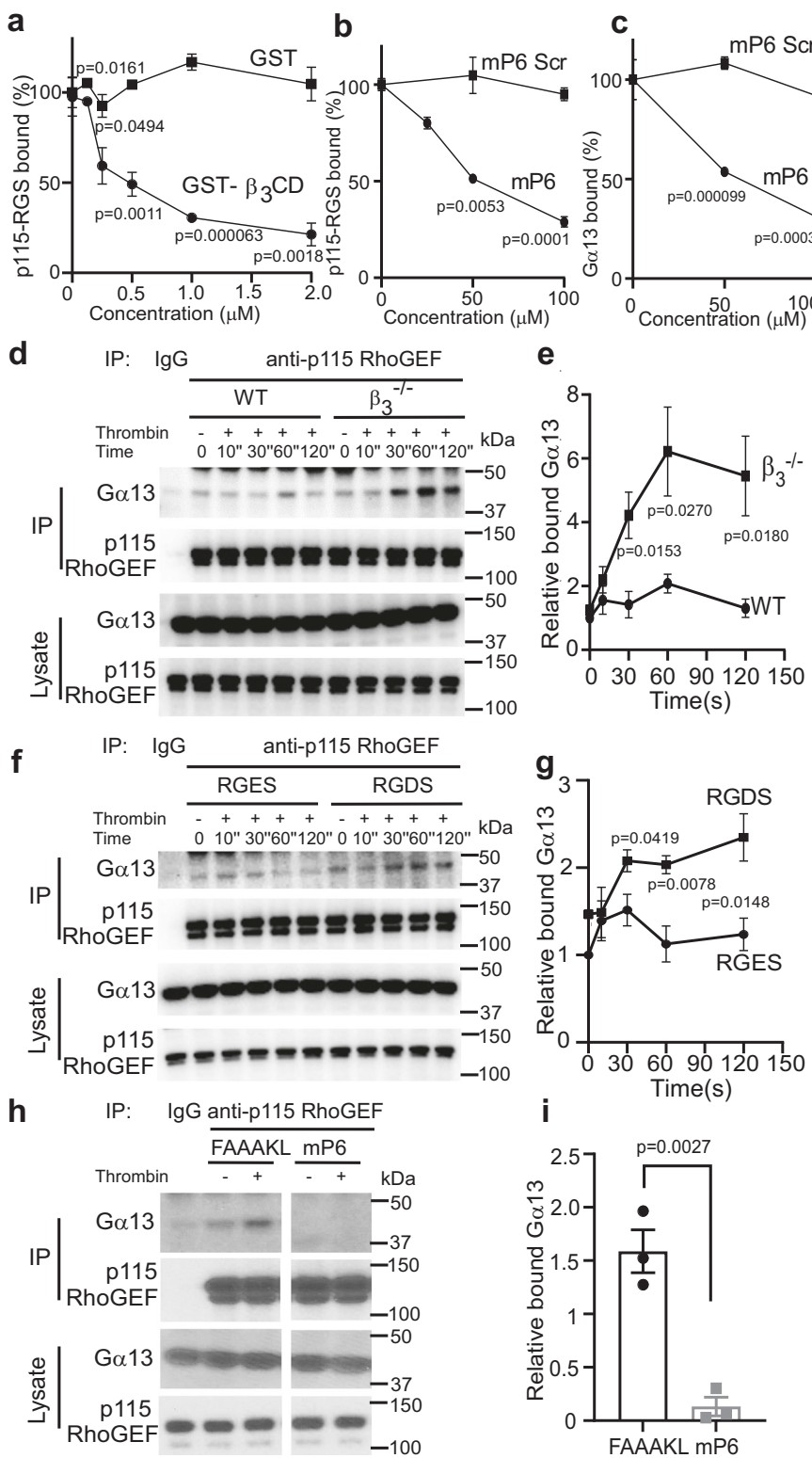

response observed in wild-type platelets. These data suggest that RhoA activity is negatively regulated by $\beta_3$. To understand whether the inhibitory role of $\beta_3$ in RhoA activation is a post-ligand binding function of $\alpha_{IIb}\beta_3$, we performed a RhoA pull-down assay in thrombin-stimulated platelets in the presence of integrin antagonist RGDS peptide or the control RGES peptide (Fig. 2c, d). RhoA activity in RGDS-treated platelets was already increased compared to the control RGES-treated platelets at the early time points following thrombin stimulation and was persistently high throughout

the course of the experiment (10 min), despite the known inhibitory effect of RGDS on platelet aggregation (see text and figures below). This contrasts with the control (RGES treated) platelets in which RhoA activation was transient (Fig. 2c, d). These results confirm that ligand binding to integrin $\alpha_{IIb}\beta_3$ greatly enhances the inhibitory effects of $\beta_3$ on RhoA activation. To further determine whether $G\alpha_{13}$ binding to the ExE motif in $\beta_3$ is responsible for the inhibitory effect of $\beta_3$, we tested the effect of mP6 peptide on thrombin-induced RhoA activation. Indeed, mP6 significantly inhibited RhoA

**Fig. 1 | Integrin β₃ cytoplasmic domain inhibits Gα₁₃-p115RhoGEF interaction in purified protein binding assays and in platelets. a** Binding of purified GST-tagged RGS domain of p115RhoGEF (GST-p115RGS) to purified Histidine-tagged Gα₁₃ (His-Gα₁₃) coated on microtiter plates is inhibited by increasing concentrations of purified GST-β₃ cytoplasmic domain fusion protein (β₃CD) compared with control GST, 3 independent experiments. **b, c** Inhibition of the binding of purified GST-p115RGS to microtiter well-coated His-Gα₁₃ (**b**) and the binding of His-Gα₁₃ to microtiter well-coated GST-p115RGS (**c**) by increasing concentration of mP6 peptide or a scrambled peptide with the same amino acid composition (mP6 Scr), 3 independent experiments. **d, e** Co-immunoprecipitation of p115RhoGEF and Gα₁₃ in WT (C57BL/6J) and β₃⁻/⁻ mouse platelets stimulated with thrombin (0.035 U/mL) for increasing lengths of time. **d** A representative Western blot; **e**, quantification of data from 4 independent experiments. **f, g** Co-immunoprecipitation of p115RhoGEF and

Gα₁₃ in resting (0) and 0.03 U/mL thrombin-stimulated WT mouse platelets pre-treated with RGDS or control RGES peptide (2 mM). **f** A representative western blot; **g** quantification of data from four independent experiments. **h, i** Co-immunoprecipitation of p115RhoGEF and Gα₁₃ in wild-type mouse platelets pre-treated with control peptide (myr-FAAAKL) or mP6 peptide (200 μM, DMSO), stimulated with thrombin (0.03 U/ mL) for 10 seconds. **h** A representative Western blot; **i** Quantification of co-immunoprecipitation of Gα₁₃. Data are from three independent experiments. Platelets used in (**d**–**g**) were pre-treated with 500 μM aspirin to exclude the differential influence of secondary TXA₂ production on Gα₁₃ pathway of WT and β₃⁻/⁻ platelets. All data are shown as mean ± SEM. Statistical analysis was done using Student's *t*-test, two tailed. Source data are provided as a Source Data file.

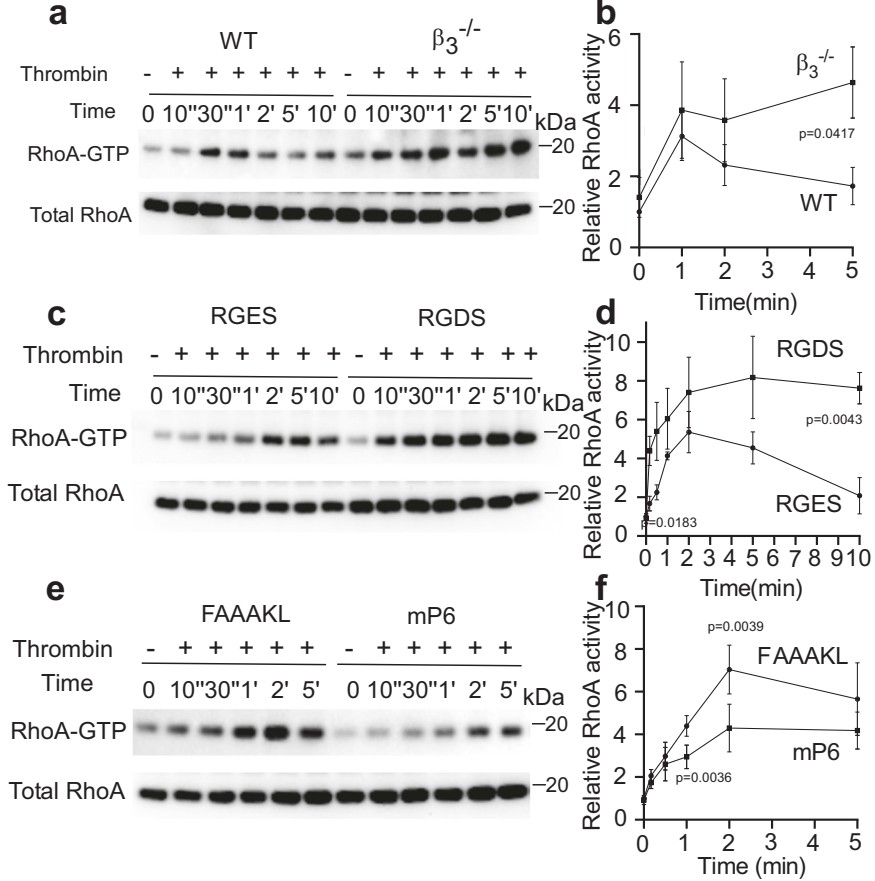

**Fig. 2 | The role of β₃ integrins in inhibiting thrombin-induced RhoA activation. a, b** Rhotekin-RBD bead pull down assay was used to analyze RhoA activation in WT or β₃⁻/⁻ mouse platelets stimulated with thrombin (0.03 U/mL) at various time points, 4 independent experiments. **c, d** RhoA pull down assay with RGDS or RGES-pre-treated (2 mM, 3 min, 22 °C) mouse platelets stimulated with thrombin (0.03 U/mL) for the indicated time points, four independent experiments. **e, f** RhoA pull down assay with 0.03 U/ mL thrombin-stimulated mouse platelets

pre-treated with control peptide or mP6 peptide (100 μM, DMSO) for 3 min at 22 °C, three independent experiments. **a, c, e** Representative western blots; **b, d, f** quantification of western blot images as analyzed using ImageJ. All data are shown as mean ± SEM. Statistical significance was determined using Student's *t* test, (**f**) paired Student's *t* test, two tailed. Source data are provided as a Source Data file.

activation as compared with the control peptide throughout the course of the experiment, even at the very early time points (Fig. 2e, f). These data indicate that Gα₁₃ binding to the ExE motif in the integrin β₃ cytoplasmic domain is likely responsible for the inhibitory effect of β₃ in Gα₁₃-mediated RhoA activation. These results further suggest that the inhibitory effect of β₃ on RhoA activation is independent of downstream integrin outside-in signaling. This is because mP6 diminishes integrin outside-in signaling[23,24] and should thus reduce the inhibitory effect of β₃ on GPCR-stimulated RhoA activation if it were dependent on outside-in signaling. Instead, mP6 mimicked the effect of β₃ to inhibit GPCR-induced

RhoA activation (Fig. 2e, f). Therefore, integrin β₃ binding to Gα₁₃ directly inhibits Gα₁₃ binding to p115RhoGEF and downstream RhoA activation in platelets, independently of outside-in signaling.

## The effect of integrin β₃ on thrombin-induced platelet granule secretion

One function of the Gα₁₃-p115RhoGEF-RhoA pathway is to stimulate granule secretion (exocytosis)[13]. To study whether and how integrin αIIbβ₃ regulates GPCR-stimulated platelet dense granule secretion, we compared thrombin-stimulated ATP secretion from wild type and β₃⁻/⁻ mouse platelets and from human platelets pre-treated with control

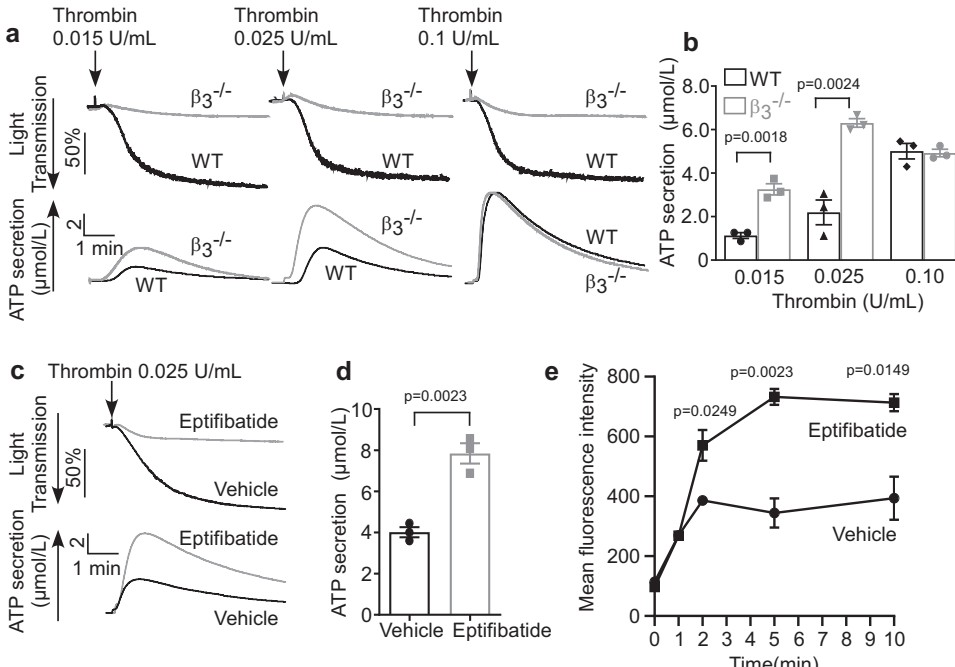

**Fig. 3 | β₃ integrins negatively regulate thrombin-induced platelet granule secretion. a** Aggregation and ATP secretion traces of washed WT and β₃⁻ᐟ⁻ platelets stimulated with thrombin. **b** Quantification of ATP secretion for **a**, three independent experiments. **c** Aggregation and ATP secretion traces of washed human platelets pre-incubated with vehicle or 10 μg/mL Eptifibatide for 3 min at 22 °C and stimulated with thrombin. **d** Quantification of ATP secretion for **c**, three independent experiments. **e** Flow cytometry analysis of P-selectin expression on mouse platelets pre-treated with control vehicle or Eptifibatide (20 μg/mL) for 3 min at 22 °C and stimulated with 0.03 U/mL thrombin at 37 °C with stirring at 1000 rpm in an aggregometer. Three independent experiments. All data are shown as mean ± SEM. Statistical significance was determined using Student's *t* test, two tailed. Source data are provided as a Source Data file.

vehicle or Eptifibatide. As expected, β₃ knockout or Eptifibatide inhibited platelet aggregation (Fig. 3a, c). However, β₃ knockout or Eptifibatide enhanced ATP secretion at low concentrations of thrombin (Fig. 3b, d), despite the expectation that abolishing platelet aggregation should also inhibit integrin-dependent platelet granule secretion. The enhancement of ATP secretion in β₃⁻ᐟ⁻ platelets was not observed at high concentrations of thrombin (Fig. 3b) because granule secretion was already maximized and could not be further increased in control platelets. This rules out the possibility that the effect of β₃ knockout was due to increased ATP concentration in platelets rather than enhanced ATP release (Fig. 3a, b). Similarly, total α-granule release of P-selectin during low-dose thrombin-stimulated platelet activation was also enhanced by Eptifibatide (Fig. 3e). These data show that integrin α$_{IIb}$β₃ negatively regulates GPCR-induced, integrin-independent granule secretion.

## Integrin β₃ negatively regulates integrin-independent secretion but stimulates integrin-dependent secretion

Platelet granule secretion can be mediated by integrin-independent and integrin-dependent signaling pathways. Thrombin stimulates a robust integrin-independent granule secretion and a relatively weak integrin-dependent secretion, which merge into one broad peak (Fig. 3c). However, a stable TXA₂ analog, U46619, can distinctively induce a relatively weak integrin-independent first-wave secretion followed by a strong integrin-dependent second wave of granule secretion[31] (Fig. 4a). Platelet aggregation stimulated by U46619 was inhibited in β₃⁻ᐟ⁻ platelets and in platelets treated with Eptifibatide (Fig. 4a, d), but the first wave of ATP secretion was enhanced in both (Fig. 4a, b, d, e). In contrast, the integrin-dependent second wave of secretion was diminished in β₃⁻ᐟ⁻ (Fig. 4a, c) and Eptifibatide-treated platelets (Fig. 4d, e). These data suggest that β₃ negatively regulates integrin-independent early wave granule secretion but stimulates integrin-dependent second wave of granule secretion.

## Integrin-independent granule secretion is RhoA-dependent

To determine whether the Gα₁₃-p115RhoGEF-RhoA pathway is involved in the integrin-independent secretion in β₃⁻ᐟ⁻ platelets, we measured U46619-induced ATP secretion in β₃⁻ᐟ⁻ platelets with or without Rho kinase inhibitor Y27632. Y27632 diminished U46619-induced granule secretion in β₃⁻ᐟ⁻ platelets (Fig. 4f), indicating that RhoA signaling is important in integrin-independent granule secretion. Previously, it was shown that RhoA also plays a key role in low-dose thrombin-induced platelet granule secretion[12,13,35].

## Gα₁₃ plays stimulatory roles in both integrin-independent and integrin-dependent granule secretion

We used Gα₁₃⁻ᐟ⁻ platelets to investigate the role of Gα₁₃ in integrin-independent and -dependent granule secretion (Fig. 5a). Platelet aggregation stimulated by thrombin was partially inhibited in Gα₁₃⁻ᐟ⁻ platelets compared with WT (Gα₁₃$^{fl/fl}$) (Fig. 5a, b), and platelet aggregation induced by high dose U46619 was only slightly affected by Gα₁₃ knockout (Fig. 5d), confirming previous reports that Gα₁₃ is not required for integrin activation and primary platelet aggregation[16,17]. Contrary to the enhancement of thrombin-induced granule secretion by β₃ knockout/inhibition, Gα₁₃⁻ᐟ⁻ platelets exhibited diminished ATP secretion after low-dose thrombin stimulation (Fig. 5b, c), indicating Gα₁₃ is needed for β₃ integrin-independent granule secretion. Furthermore, both the first wave and second wave of ATP secretion induced by U46619 were reduced in Gα₁₃⁻ᐟ⁻ platelets (Fig. 5d, e). This contrasts with the effects of β₃ knockout or inhibition, which selectively inhibited integrin-dependent second-wave granule secretion, but enhanced integrin-independent first wave secretion (Fig. 4a–e). This marked difference between Gα₁₃ knockout and β₃ integrin knockout/inhibition shows that Gα₁₃ mediates both the GPCR-induced integrin-independent granule secretion via the Gα₁₃-p115RhoGEF-RhoA pathway and integrin-dependent granule secretion via its role in mediating integrin outside-in signaling.

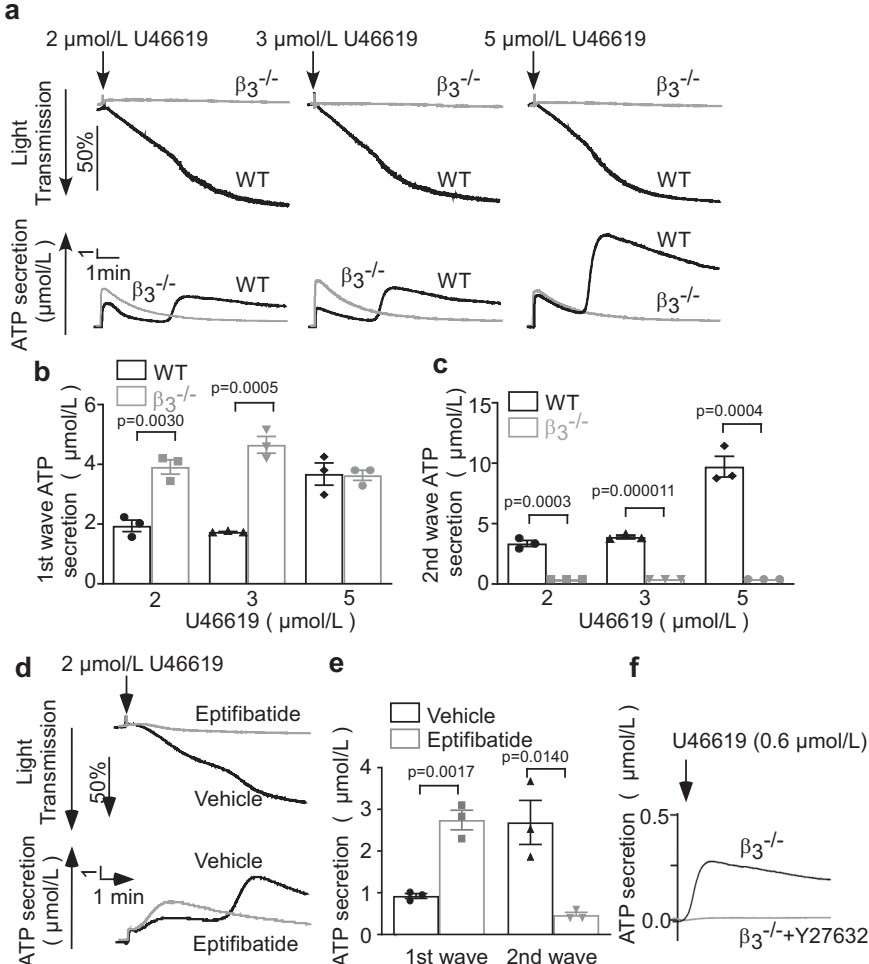

**Fig. 4 | β₃ integrins negatively regulate integrin-independent platelet secretion but stimulate integrin-dependent secretion. a** Aggregation and ATP secretion traces of washed WT and β₃⁻/⁻ platelets induced by U46619 at 37 °C. Note the two waves of ATP secretion in WT platelets and a higher single (first) secretion wave in β₃⁻/⁻ platelets. **b, c** Quantification of the first (**b**) and second (**c**) waves of secretion as shown in **a**, three independent experiments. **d** Aggregation and ATP secretion traces of washed human platelets pre-incubated with vehicle or 10 µg/mL

Eptifibatide for 3 min at 22 °C followed by stimulation with U46619 at 37 °C. **e** Quantification of secretion data from (**d**), 3 independent experiments. **f** U46619-induced ATP secretion traces of β₃⁻/⁻ platelets pre-incubated with control vehicle or 10 µM Y27632 for 3 min at 22 °C. All data are shown as mean ± SEM. Statistical significance was determined using Student's *t*-test, two tailed. Source data are provided as a Source Data file.

## The ExE motif peptide mP6 inhibits both integrin-independent and integrin-dependent granule secretion

The mP6 peptide is a mimic of the β₃ cytoplasmic ExE motif that binds to Gα₁₃ to inhibit both the Gα₁₃-p115RhoGEF interaction (Fig. 1) and the Gα₁₃-β₃ interaction[23,36]. We thus hypothesized that mP6 should inhibit both integrin-independent and integrin-dependent secretion. Indeed, mP6 inhibited low-dose thrombin-induced ATP secretion (Fig. 6a, b) and diminished both integrin-independent and dependent waves of ATP secretion (Fig. 6c, d) induced by thrombin and U46619. Furthermore, in the presence of extracellular fibrinogen, mP6 selectively reduced secretion-dependent secondary platelet aggregation without affecting primary aggregation (Supplementary Fig. 4). mP6 also inhibited low-dose thrombin-induced dense granule secretion of serotonin (Fig. 6e), α granule secretion of β-TG (Supplementary Fig. 5a), and surface expression of α granule membrane protein P-selectin (Fig. 6f) in platelets. Thus, dual blockage of both Gα₁₃-p115RhoGEF and Gα₁₃-integrin interaction by mP6 results in potent inhibition of both integrin-independent and dependent secretion of dense granules and α granules, but minimally affects primary platelet aggregation.

## The effect of ExE motif peptide on platelet granule secretion in vivo during myocardial ischemia/reperfusion injury in mice

Platelet granules contain high levels of platelet agonists, pro-coagulation molecules, pro-inflammatory cytokines and receptors, and growth factors[28]. Secretion of granule contents induced via activation of GPCRs and other platelet receptors results in not only great amplification of thrombosis, but also inflammation, cell proliferation, and atherosclerosis[28]. In this respect, the potent inhibition of platelet granule secretion and secretion-dependent secondary platelet aggregation as shown here provides an important mechanism underlying the strong anti-thrombotic effect of mP6 and the modified version (M3mP6) engineered into lipid-stabilized, high-loading peptide nanoparticles (HLPN) in vitro and in vivo[17,36]. However, it does not entirely explain why M3mP6 also protects the heart from myocardial ischemia–reperfusion injury (MI/RI) in a mouse model[36], as current clinically used anti-thrombotic drugs are not effective in treating MI/RI[37]. MI/RI is a grave thrombo-inflammatory condition responsible for post-ischemia cardiac injury and heart failure, in which inflammation has been suggested to play an important role in addition to microvascular thrombosis[38,39]. Interestingly, secretion of serotonin from platelet granules was recently found to cause neutrophil degranulation

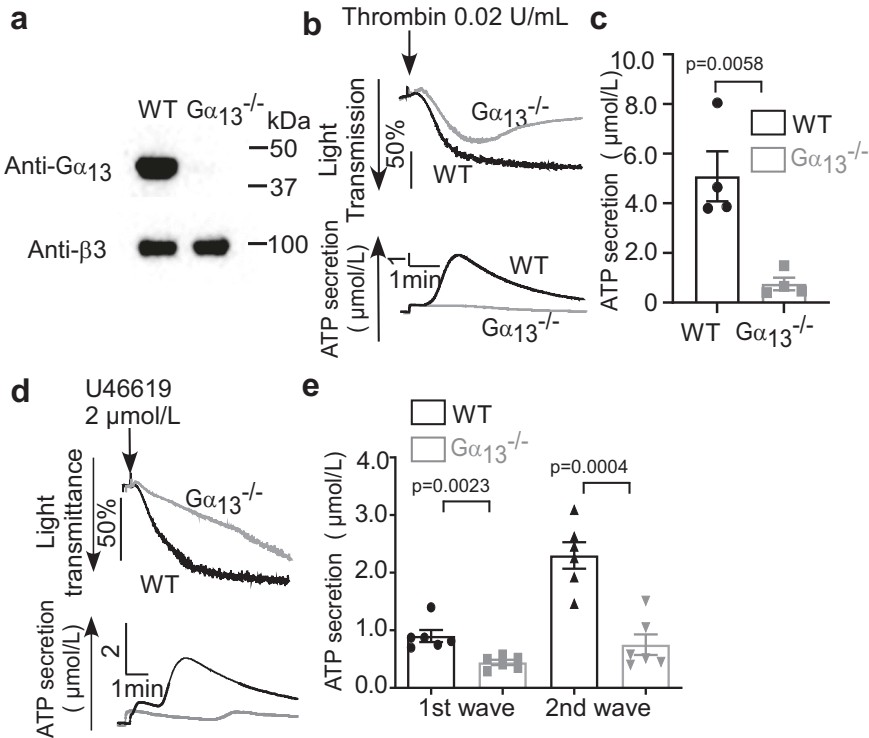

**Fig. 5 | Gα₁₃ stimulates both integrin-independent and integrin-dependent granule secretion. a** Washed WT(Gα₁₃$^{fl/fl}$) and Gα₁₃$^{-/-}$ mouse platelets were solubilized and immunoblotted with antibodies specifically recognizing mouse Gα₁₃, and integrin β₃. **b** Aggregation and ATP secretion traces of washed WT and Gα₁₃$^{-/-}$ platelets stimulated with thrombin. **c** Quantification of secretion as shown in **b**, four independent experiments. **d** Aggregation and ATP secretion traces of WT and Gα₁₃$^{-/-}$ platelets stimulated with 2 μmol/L U46619. **e** Quantification of ATP secretion as shown in **d**, six independent experiments. Aggregation assays were performed without exogenously added fibrinogen. All data are shown as mean ± SEM. Data were analyzed using Student's *t*-test, (**e**) paired Student's *t*-test, two tailed. Source data are provided as a Source Data file.

and exacerbation of MI/RI in mouse models and in patients with acute coronary syndrome[30], which is consistent with the observation that M3mP6 HLPN also inhibit neutrophil function during MI/RI[36]. Thus, we used the mouse left anterior descending (LAD) coronary artery ligation-reperfusion model to test whether blocking the Gα₁₃-p115RhoGEF and Gα₁₃-β₃ interactions using M3mP6 HLPN would effectively inhibit serotonin release in vivo during myocardial ischemia-reperfusion. Indeed, M3mP6 HLPN significantly reduced plasma serotonin levels (Fig. 6g) and plasma levels of platelet α−granule proinflammatory chemokine protein β-TG (Supplementary Fig. 5b) in post-MI/R mice. Thus, in addition to its potent antithrombotic effects, blocking platelet secretion of serotonin and other anti-inflammatory factors enriched in platelet granules provides another key mechanism explaining the ability of M3mP6 to reduce cardiac injury and improve heart function and survival after MI/RI in mice[36]. Importantly, M3mP6 HLPN do not affect hemostasis nor increase bleeding risk[36], despite the considerable anti-thrombotic/anti-inflammatory effects of M3mP6 resulting from inhibition of platelet secretion. These properties are uniquely advantageous over current integrin antagonists and other platelet inhibitors that primarily inhibit platelet adhesion/aggregation, leading to an enhanced risk for the severe adverse effect of bleeding.

## Discussion

In this study, we discovered that integrin β₃, by binding to Gα₁₃, directly inhibits the binding of p115-RhoGEF to Gα₁₃, and negatively regulates the GPCR-induced Gα₁₃-p115RhoGEF-RhoA signaling pathway, demonstrating the direct cross-talk by which integrins regulate GPCR signaling. We further show that β₃ integrins, by binding to Gα₁₃, play dual roles of (1) negatively regulating Gα₁₃-mediated, integrin-independent granule secretion and (2) transmitting integrin outside-in

signaling to stimulate integrin-dependent granule secretion. Furthermore, we show that the ExE motif peptide mP6, derived from the sequence of the Gα₁₃-binding site on β₃, inhibits both Gα₁₃-p115Rho-GEF-RhoA-mediated integrin-independent granule secretion and integrin-dependent outside-in signaling-mediated granule secretion without compromising primary platelet aggregation. These data demonstrate an approach to robustly inhibit platelet granule secretion without compromising primary platelet thrombus formation and hemostasis (Fig. 6h).

The discovery of crosstalk between integrin and GPCR pathways to inhibit GPCR-mediated RhoA activation has significant implications in our understanding of how GPCRs and integrins may coordinate cell adhesion, spreading, retraction, and migration as well as exocytosis. An important role of GPCR signaling is to regulate the functions of the major adhesion receptor integrins. GPCR-coupled Gα$_q$ and Gα$_i$ pathways induce Rap1/talin/kindlin-dependent inside-out signaling, activating the ligand binding functions of integrins, and thus integrin-dependent cell adhesion/aggregation (Supplementary Fig. 1)[40–43]. GPCRs also stimulate Gα₁₃-dependent RhoA activation important in cell retraction. Previous studies indicated that integrin-mediated cell spreading requires inhibition of RhoA-mediated cell retraction, and that coordinated cell spreading, and retraction drives cell migration[8,24,44]. Our finding that ligand-bound integrins may directly inhibit the Gα₁₃-mediated RhoA activation pathway suggests a potential mechanism by which GPCR and integrins coordinate to stimulate cell spreading and inhibit cell retraction, only at locations where integrin-mediated adhesion occurs but allows cell retraction where integrins are not ligated, thus controlling the direction of cell movement.

In previous studies, we reported that Gα₁₃ binding to the cytoplasmic domain of β₃ integrin subunits mediate integrin outside-in

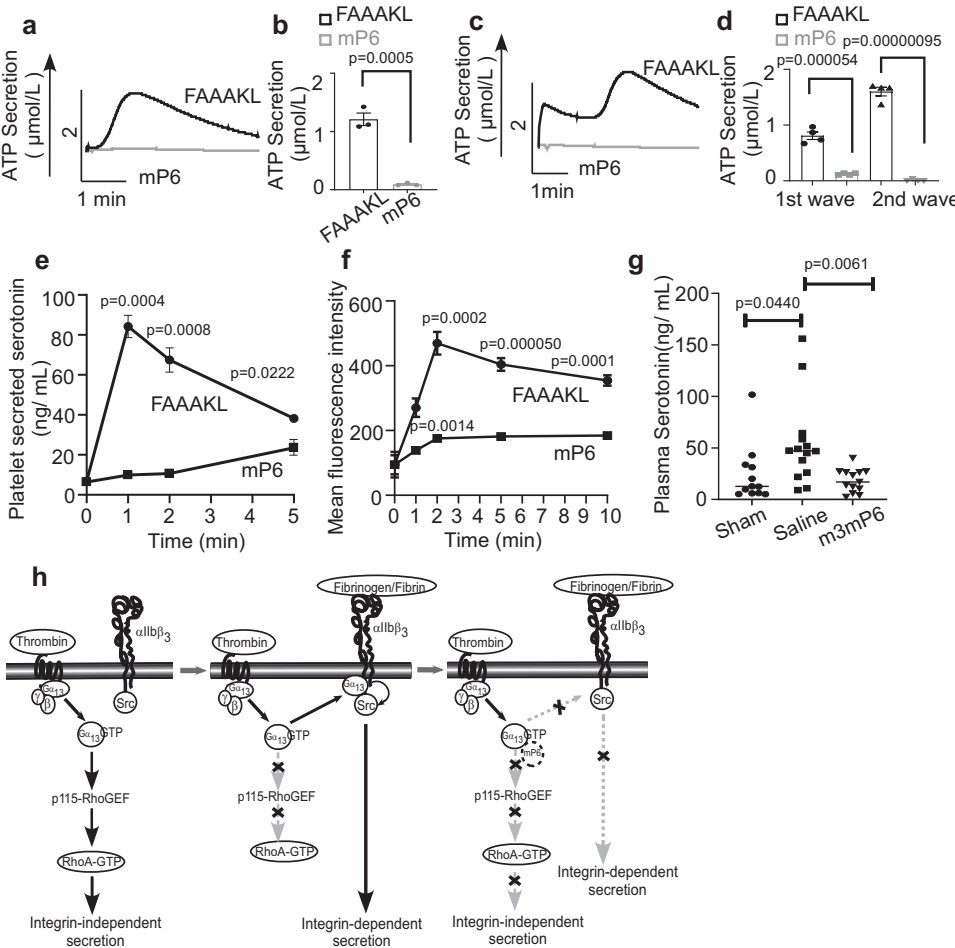

**Fig. 6 | Gα13 antagonistic peptide mP6, inhibits both integrin-independent and integrin-dependent granule secretion. a, c** ATP secretion traces of washed mouse platelets pre-incubated with 20 μM control FAAAKL or mP6 high loading peptide nanoparticles (HLPNs) and then stimulated with 0.03 U/mL thrombin (**a**) or 3 μmol/L U46619 (**c**). **b, d** Quantification of ATP secretion as shown in **a**, n = 3 or (**c**), n = 4, independent experiments. **e, f** Effects of control FAAAKL or mP6 HLPNs (20 μM) on serotonin secretion (**e**), or on P-selectin expression (**f**), in mouse platelets stimulated with 0.03U/ mL thrombin in an aggregometer stirring at 37 °C and 1000 rpm. n = 3, independent experiments. The assays were performed using washed platelets without adding fibrinogen. **g** Plasma serotonin levels in sham, saline, and M3mP6 HLPN treatment groups 24 h after myocardial infarction with reperfusion. Sham group, n = 12; Saline group, n = 14; M3mP6 group, n = 13, independent animals. Data in **b, d–f** are shown as mean ± SEM and in **g** are shown as median ± SEM. Statistical significance was determined using Student's t-test, two tailed. **h** A schematic of crosstalk between the (G protein-coupled receptor) GPCR-coupled Gα13 pathway and integrin αIIbβ3 in regulating RhoA activation and platelet granule secretion. Left panel: GPCR agonists such as thrombin induce Gα13-p115RhoGEF-dependent activation of RhoA, which mediates integrin-independent granule secretion. Center panel: following integrin ligation, Gα13 interacts with the β3 cytoplasmic domain ExE motif, which inhibits Gα13 binding to p115RhoGEF, resulting in inhibition of p115RhoGEF-mediated RhoA activation and integrin-independent granule secretion. Gα13 binding to β3, however, also mediates integrin outside-in signaling leading to Src activation and integrin-dependent granule secretion. Right Panel: The β3-derived mP6 peptide binds to Gα13 to inhibit Gα13 interacting with both p115-RhoGEF and β3. Thus, mP6 potently inhibits both integrin-independent and integrin-dependent secretion without affecting integrin ligation and primary platelet adhesion/aggregation necessary for normal hemostasis. Source data are provided as a Source Data file.

signaling leading to cell spreading[17,24,44]. We also demonstrated that cell spreading induced by Gα13-dependent integrin outside-in signaling requires transient inhibition of RhoA[17,23]. The mechanism by which outside-in signaling mediates transient RhoA inhibition has not been clear, but there is evidence suggesting a potential role of integrin/Src-dependent p190RhoGAP activation[45] and p190RhoGAP-mediated RhoA inactivation[46,47]. This mechanism was demonstrated in the experimental system of integrin-dependent cell spreading in the absence of GPCR agonist stimulation, and thus is a likely mechanism of inhibiting integrin-dependent RhoA activation, which occurs in the late retraction phase of integrin outside-in signaling following calpain-mediated abolishment of c-Src binding to β3[48]. Here we show that the binding of an ExE motif in the cytoplasmic domain of β3 integrin to Gα13 directly inhibits Gα13 binding to p115RhoGEF and thus inhibits GPCR-mediated RhoA activation independently of the outside-in signaling pathway. Therefore, we demonstrate that integrins inhibit RhoA signaling by both direct inhibition of GPCR/Gα13-mediated RhoA activation and facilitating deactivation of RhoA via the Src-dependent early phase outside-in signaling pathway.

We demonstrate here that the cross-talk between integrin and GPCR pathways plays an important role in regulating exocytosis (granule secretion) in platelets. Platelet granule secretion can be stimulated by GPCR agonists, independent of integrins, which releases integrin ligands and other cargo molecules to facilitate platelet activation, adhesion, and aggregation. GPCR-stimulated granule secretion involves Gαq, Gαi, and Gα13 and their downstream signaling pathways, like phospholipase C (PLC) β-dependent calcium elevation and protein kinase C activation, the phosphoinositide 3-kinase-Akt pathway, and the p115-RhoGEF-RhoA pathway (Supplementary Fig. 1). Gαq and Gαi pathways are also critical in inside-out signaling leading to integrin activation. Ligand binding to integrins not only mediates platelet adhesion and aggregation, but also induces integrin-dependent

granule secretion, critical for the second wave of platelet aggregation, thrombus expansion, and stabilization[31]. Here we show that ligand binding to integrin $\alpha_{IIb}\beta_3$ negatively regulates GPCR-stimulated, integrin-independent granule secretion by direct inhibition of the $G\alpha_{13}$-p115RhoGEF-RhoA pathway. In contrast, integrin-dependent granule secretion is mediated via the $G\alpha_{13}$-dependent integrin outside-in signaling pathway, which activates the PI3K-Akt3 signaling pathway and activation of immunoreceptor tyrosine-based activation motif (ITAM) signaling and thus PLCγ[26,49,50]. The finding that $\beta_3$ integrins negatively regulate integrin-independent secretion while stimulating integrin-dependent granule secretion is likely to be physiologically relevant as it minimizes granule secretion in circulating platelets, even if stimulated by low concentrations of agonists, but greatly enhances platelet granule secretion only when platelets adhere and aggregate to form a thrombus. Importantly, platelet granules contain large amounts of growth factors and pro-inflammatory cytokines, which are known to stimulate inflammation, migration, and proliferation of vascular cells including endothelial cells and smooth muscle cells. Thus, platelet granule secretion promotes inflammation, atherosclerosis, and angiogenesis. However, patients with Glanzmann Thrombasthenia (with lack of expression or function of $\alpha_{IIb}\beta_3$)[51] or $\beta_3$ knockout mice[52] were not protected but paradoxically showed enhanced development of atherosclerosis. These observations are not consistent with the current assumption that inhibition of $\beta_3$-dependent platelet adhesion and secretion would reduce atherogenesis. Our finding that $\beta_3$ knockout enhances integrin-independent exocytosis in platelets provides a potential mechanism for this pro-atherogenic effect. Thus, it would be interesting to investigate whether selectively targeting $G\alpha_{13}$-integrin interaction would help prevent/reduce atherosclerosis and other vascular inflammatory conditions.

Most current anti-platelet drugs have inhibitory effects on granule secretion, but robustly inhibit platelet adhesion/ aggregation and have adverse effects of bleeding and vascular leakage, which may exacerbate inflammation. Additionally, integrin antagonists, as shown in this study, enhance integrin-independent granule secretion, even though they inhibit integrin-dependent granule secretion. In contrast, we show that the inhibitor peptide derived from the $G\alpha_{13}$ binding sequence of integrin $\beta_3$, mP6, has the dual effects of inhibiting both the $G\alpha_{13}$-p115RhoGEF-RhoA pathway and integrin outside-in signaling, and thus inhibits both integrin-independent and dependent granule secretion as well as the second wave of platelet aggregation without affecting primary platelet adhesion and aggregation. This peptide also showed no effect on hemostasis in vivo[23]. This represents a new approach to effectively inhibit granule secretion both in and out of the localized thrombus without impairing hemostatic thrombus formation. Furthermore, the recent finding that platelet secretion of serotonin promotes the role of neutrophils in causing cardiac ischemia/ reperfusion injury, provides a mechanism for the beneficial effects of inhibitors of $G\alpha_{13}$-$\beta_3$ integrin interaction in treating cardiac ischemia/ reperfusion injury[30,36].

In summary, our data suggest a new paradigm for cross-talk between GPCR and integrin signaling in which integrins directly inhibit $G\alpha_{13}$-mediated GPCR signaling (Fig. 6h). We further demonstrate a unique strategy of targeting this G protein-integrin crosstalk to strongly inhibit both the GPCR- and integrin-mediated granule secretion in platelets without affecting the primary adhesive function of integrins (Fig. 6h). We also provide proof-of-concept in vivo data supporting a novel approach to target integrin-$G\alpha_{13}$ crosstalk for the treatment of secretion-mediated thrombo-inflammatory conditions such as MI/RI. This contrasts with the lack of effectiveness of current anti-thrombotic (anti-platelet) therapies in treating MI/RI. In addition, current anti-platelet drugs strongly inhibit the adhesive function of platelet integrins, thus causing exacerbation of hemorrhage and vascular leakage, which is detrimental in the treatment of thrombo-inflammation. Our new approach and the underlying mechanisms

uncovered in this study avoid this problem and suggest a promising future direction in treating MI/RI and possibly other thrombo-inflammatory diseases.

## Methods

### Ethics statement
All animal studies were approved by the Institutional Animal Care Committee of the University of Illinois at Chicago (animal protocol number: 22-184). For studies using human platelets, Institutional Review Board approval (1999-0610) was obtained from the University of Illinois at Chicago, and informed consent was obtained from all healthy donors in accordance with the Declaration of Helsinki.

### Regents and materials
Myristoylated peptides, mP6 (Myr-FEEERA), mP6Scr (Myr-ERAFEE), M3mP6 (Myr-FEEERL), and control peptide (Myr-FAAAKL) were synthesized and purified by the Research Resources Center of the University of Illinois at Chicago, CPC Scientific Inc. (San Jose, CA) or Peptide 2.0 Inc. (Chantilly, VA). These peptides were encapsulated into high-loading peptide nanoparticles (HLPN) to allow efficient entry into cells[36]. RGDS and RGES peptides were synthesized by Peptide 2.0 Inc. (Chantilly, VA). Apyrase, prostaglandin E1 (PGE1) and aspirin were from Sigma-Aldrich (St Louis, MO). α-Thrombin and fibrinogen were from Enzyme Research Laboratories (South Bend, IN). Luciferase/luciferin reagent was from Chrono-log (Havertown, PA). U46619 and Y27632 were from Calbiochem. Anti-$G\alpha_{13}$ (67188-1-lg, 1:1000 for western blot), anti-p115 RhoGEF (11363-1-AP, 2 μg per 500 μL cell lysate), anti-GAPDH (60004-1-lg, 1:50,000 for western blot) and anti-integrin $\beta_3$ (18309-1-AP, 1:1000 for western blot) were from Proteintech (Rosemont, IL). Rabbit IgG (2729S, 2 μg per 500 μL cell lysate) and anti-p115 RhoGEF (D25D2, 1:1000 for western blot) were from Cell Signaling technology Inc. (Danvers, MA). Protein A/G PLUS-Agarose (sc-2003) beads were from Santa Cruz Biotechnology Inc (Dallas, TX). FITC-conjugated rat anti-mouse CD62P antibody (553744, 1 μg/ $10^5$ platelets) and FITC-conjugated rat IgG1 λ isotype control (553995, 1 μg/ $10^5$ platelets) were from BD Pharmingen. Eptifibatide was from Merck & Co., Inc. (White house station, NJ). RhoA activation assay Biochem Kit (BK036) was from Cytoskeleton, Inc (Denver, CO). Serotonin Research ELISA kit (BA-E-5900R) was from LDN Labor Diognostika (Nordhorn, Germany). Mouse beta-TG (beta-Thromboglobulin) ELISA Kit (MBS2510762) was from MyBiosource, Inc (San Diego, CA). K2EDTA tubes (367842) were from BD Vacutainer (Boston, MA).

### Animals
$G\alpha_{13}^{flox/flox}$ mice were generous gifts from Dr. Stefan Offermanns, Max Planck Institute for Heart and Lung Research (Bad Nauheim, Germany)[16]. Generation of platelet-specific $G\alpha_{13}$ knockout ($G\alpha_{13}^{fl/fl}$, PF4-Cre$^+$, referred to as $G\alpha_{13}^{-/-}$) mice were previously described[25,53]. Integrin $\beta_3$ knockout ($\beta_3^{-/-}$) mice were obtained from Jackson Laboratory (Bar Harbor, ME) and were maintained on a C57BL/6J background. C57BL/6J and $G\alpha_{13}^{fl/fl}$ (referred to as $G\alpha_{13}^{+/+}$) mice were used as their respective controls. Mice used in this study were 8 to 16-weeks-old with an equal sex ratio. Animals were housed and bred in the Biologic Resources Laboratory at the University of Illinois at Chicago under 12 h light-dark cycles, controlled temperature (-23°) and 40–50% humidity with free access to food and water. A randomized approach of choosing mice was used throughout the study, using all mice with the correct genotype without bias.

### Preparation of platelets
Washed mouse and human platelets were prepared as previously described[17,23,26]. Fresh mouse blood was drawn from the inferior vena cava and added with one-ninth volume of anti-coagulant solution (85 mM trisodium citrate, 83 mM D-glucose, and 21 mM citric acid). After adding a final concentration of 0.1 μg/mL prostaglandin E1 and

1 U/mL apyrase, platelet-rich plasma (PRP) was isolated by centrifugation of whole blood at 200×*g* at 22 °C for 10 min, and mouse platelets were further isolated by centrifugation at 870×*g* at 22 °C for 10 min. In some experiments, an additional 1 U/ mL apyrase and 1/100 volume of 0.5 M EDTA (pH 8.0) was added to PRP before centrifugation to minimize platelet activation. For human blood, a one-seventh volume of ACD (85 mM trisodium citrate, 110 mM D-glucose, and 78 mM citric acid) was used as anticoagulant and centrifuged at 300 g for 20 min, 22 °C, to obtain PRP, which was further centrifuged at 700×*g* at 22 °C for 10 min to isolate platelets. Thereafter, mouse or human platelets were washed once with CGS buffer (sodium chloride 0.12 M, D-glucose, 0.03 M, trisodium citrate 0.0129 M, pH 6.5) and resuspended in modified Tyrode's buffer (12 mM NaHCO$_3$, 138 mM NaCl, 5.5 mM glucose, 2.9 mM KCl, 0.42 mM NaH$_2$PO$_4$, 10 mM *N*-2-hydroxyethylpiperazine-*N'*−2-ethanesulfonic acid, 1 mM MgCl$_2$, pH 7.4). Washed platelets were then allowed to rest at room temperature for at least 1 h. CaCl$_2$ (1 mM) was added to platelet suspensions 10 minutes before use.

## Platelet aggregation and granule secretion

Platelet aggregation and dense granule secretion were measured in a turbidimetric platelet lumi-aggregometer (Chrono-log) at 37 °C with stirring (1000 rpm), and the real-time traces were recorded using Aggro/Link8 (version 1.2.9) from CHRONO-LOG CORP[25,54,55]. Peptides, (mP6 or control peptide), encapsulated into HLPN[36], were added to the platelet suspension for 3 minutes at room temperature prior to assay. Platelet dense granule secretion of ATP was assessed by using luciferin-luciferase reagent in the aggregometer[31,56]. Quantification was performed using an ATP standard curve. Mouse platelet secretion of serotonin from dense granules was assessed by a Serotonin Research ELISA kit. The total α granule of β-TG secretion was assessed by mouse β-TG ELISA kit. Specifically, platelet suspensions (with or without agonist stimulation) were placed on ice to stop the secretion and were immediately spun down at 10,000×*g* at 4 °C, for 10 min. Supernatants were saved for ELISA analysis following manufacturer's instructions.

Platelet α-granule secretion was also analyzed by estimating the surface expression of P-selectin[57]. Mouse platelets isolated from freshly drawn blood as described above were washed in CGS buffer (with 0.1% BSA, pH6.5) and resuspended in modified Tyrode's buffer (with 0.1% BSA, pH 7.4) containing 1 mM CaCl$_2$ and 1 mM MgCl$_2$. Washed platelets in Tyrode's buffer were pre-incubated with control vehicle or 20 μg/mL Eptifibatide, 20 μM control peptide or mP6 HPLNs for 3 min at room temperature, and then stimulated with thrombin in an aggregometer at 37 °C at 1000 rpm stirring rate for the indicated time points. After fixing with 2% paraformaldehyde, platelets were incubated with an FITC-conjugated rat anti-mouse P-selectin antibody or rat IgG control for 30 min at 22 °C in the dark. After dilution 10-fold in PBS (containing 1% BSA), platelet P-selectin expression was analyzed using a Accuri C6 flow cytometer with CFlow Plus software (version 1.0.227.4) (BD Biosciences). The Gating strategy for flow cytometry analysis is shown in Supplementary Fig. 6.

## Co-immunoprecipitation

Washed mouse platelets resuspended in modified Tyrode's buffer (3 × 10$^8$ /mL) were stimulated with thrombin for the indicated time points in an aggregometer (37 °C, 1000 rpm), and then were solubilized in HEPES-NP40 Buffer (20 mM HEPES, 10 mM MgCl$_2$, 150 mM NaCl, 1% NP-40, 1 mM EGTA, 10% Glycerol, 1 mM sodium orthovanadate, 1 mM NaF, pH 7.4), with complete protease inhibitor cocktail (1 tablet/10 ml buffer, Roche Diagnostics, Mannheim), on ice for 30 min. Platelet lysates were then cleared at 14,000×*g* for 10 min at 4 °C and the supernatants incubated with 4 μg/ml of anti-p115RhoGEF (11363-1-AP) or IgG over night at 4 °C. They were then further incubated with Protein A/G PLUS-Agarose beads, rotating at 4 °C for 2 h. After 3 washes with lysis buffer, immunoprecipitants were analyzed by SDS-polyacrylamide gel electrophoresis and immunoblotted with antibodies against p115-RhoGEF (D25D2) or Gα$_{13}$. When comparing Gα$_{13}$-p115 RhoGEF interactions in WT (C57BL/6 J) or β$_3^{-/-}$ platelets, and in RGES or RGDS (2 mM) pre-treated platelets[17], and in control vehicle or 40μM Eptifibatide pre-treated platelets, 500 μM aspirin[31] was also added to platelets to block and exclude the differential influence of secondary TXA$_2$ production on the Gα$_{13}$ pathway. In some experiments, a control peptide or mP6 peptide (200 μM in DMSO) was used to pretreat platelets for 3 minutes prior to assay. Quantitation of western blots was performed using ImageJ. The data were normalized relative to the values obtained from platelets in the resting state (0 min).

## GTP-RhoA pull down assay

For the GST-RhoA pull down assay, all necessary regents including lysis buffer, washing buffer, Rhotekin-RBD protein beads, protease inhibitor cocktail, and anti-RhoA monoclonal antibody were those included in the RhoA activation assay Biochem Kit. Briefly, isolated mouse platelets (3 × 10$^8$/mL) resuspended in modified Tyrode's buffer were lysed quickly with RhoA lysis buffer (added with 1× protease inhibitor cocktail) after thrombin stimulation for the indicated time points in an aggregometer. Lysates were cleared by centrifugation at 17,000×*g* for 2 min at 4 °C, and the supernatants were incubated for 1 h with 50 μg Rhotekin-RBD protein beads/sample. Samples were washed once using washing buffer and then immunoblotted with anti-RhoA antibody (ARH05, 1:500 for western blot). Cell lysates were also immunoblotted with anti-RhoA as loading control. In some experiments, integrin inhibitor RGDS or control RGES (2 mM), and mP6 or control peptide (100 μM in DMSO) were used to treat platelets for 3 minutes before thrombin stimulation for the indicated time points. Quantitation of western blots was performed using ImageJ. The data are normalized relative to the values obtained from platelets in the resting state (0 min).

## In vitro competition assay

GST-β$_3$CD was purified as described before[17,23]. For GST-β$_3$CD cDNA construction, integrin β$_3$ cytoplasmic domain (716-762) cDNA was amplified by PCR and cloned into pGEX-4T2 vector using Bam HI and Xho I restriction sites. The forward primer is 5′-CGTGGATC-CAAACTCCTCATCACCATCCACGACC-3′; the reverse primer is 5′-GCGCTCGAGTTAAGTGCCCCGGTACGTGATATTG-3′. GST-β$_3$CD was purified from BL21 (DE3) *E. coli* with IPTG induction and using glutathione-conjugated beads[17]. Histidine-tagged Gα$_{13}$(His-Gα$_{13}$) was previously provided by Dr. Tohru Kozasa (University of Illinois at Chicago, Chicago, IL)[58]. GST-RGS was purified from BL21 cells transformed with vector PGEXKG p115-RGS. His-Gα$_{13}$ (5 μg/mL) or 2.5 μg/mL GST-RGS were coated onto 96-well microtiter plates overnight. After blocking with 5% BSA overnight at 4 °C, plates were washed three times with PBS. Increasing concentrations of purified GST-β$_3$CD protein/ GST control protein, or inhibitor peptide mP6 or control peptide were then added to the plates in Tris-NP40 buffer (50 mM Tris, 10 mM MgCl$_2$, 150 mM NaCl, 1% NP-40)[23]. After incubation for 1 h at room temperature, 0.25 μg/mL GST-RGS or 2.5 μg/mL His-tagged recombinant Gα$_{13}$ was added to the plates for another 2 h. Bound RGS of p115 or Gα$_{13}$ was estimated with anti-p115 RhoGEF (D25D2) or anti-Gα$_{13}$ antibodies, anti-rabbit, or anti-mouse HRP-conjugated secondary antibodies and 3,3′,5,5′-tetramethylbenzidine substrates (Thermo Fisher Scientific Pierce Biotechnology, Rockford, IL). The wells were washed three times with PBST between each step. The reactions were terminated with 0.16 M sulfuric acid and measured for OD$_{450nm}$ with a FlexStation3 Multi-mode Microplate reader (Molecular Devices, LLC).

## Mouse myocardial ischemia and reperfusion model

The MI/RI model was performed as previously described[36]. C57BL/6J mice were randomly assigned to sham and surgical groups. Following

induction of ischemia, M3mP6 HLPN or saline was injected at a bolus dose of 5 μmol/kg through the jugular vein cannula. After 1 h of ischemia, the suture was cut and removed, followed by M3mP6 HLPN (2.5 μmol/kg/h) or saline infusion through the jugular vein cannula for 24 h. To obtain plasma, blood was drawn via the vena cava into $K_2$EDTA tubes and centrifuged at 2000 g within 30 min of collection. Plasma serotonin[30] and β-TG were quantified using ELISA kits according to manufacturer's instructions.

### Statistical analysis

Data are expressed as means (or median) ± SEM. For parametric data, differences between groups of samples were evaluated with Student's $t$-test, with GraphPad Prism software. For nonparametric data, statistical significance was determined using the Mann–Whitney test. A $p$-value ≤ 0.05 was considered statistically significant.

### Reporting summary

Further information on research design is available in the Nature Portfolio Reporting Summary linked to this article.

## Data availability

The authors declare that the data generated in this study are provided in the Supplementary Information/Source Data file. Additional information can be obtained from the corresponding author upon reasonable request. Source data are provided with this paper.

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

## Acknowledgements

We thank Dr. Stefan Offermanns, Max Planck Institute for Heart and Lung Research, Bad Nauheim, Germany, for providing Gα$_{13}$ knockout mice. This work is supported in part by grants from the National Heart, Lung, and Blood Institute grants R35HL150797 (X.D.), RO1HL080264 (X.D), RO1HL125356 (X.D.), R44HL156560-01A1 (R.A.S.), and R44 HL142396 (R.A.S), and by the Lilly Research Award Program (G.D. and X.D.).

## Author contributions

Y.Z., X.Z. and B.S. contributed equally to experimentation, data analysis, and manuscript writing. Y.B., C.C., A.S., C.W. and A.M. performed some experiments and data analysis. G.D. contributes to important discussions and editing. R.A.S. contributed to experimental planning, analysis, and manuscript writing, and N.C. contributed to some experiments, technical supervision, data analysis, and manuscript writing. X.D. led the project, designed research, analyzed/interpreted data, and wrote the manuscript.

## Competing interests

University of Illinois at Chicago holds patents related to this study. X.D. holds equity interests in DMT, Inc., which licenses UIC technology. The remaining authors declare no competing interests.
