## [Peer review file · Nature Communications]

REVIEWER COMMENTS

Reviewer #1 (Remarks to the Author):

This is an interesting manuscript on the mechanisms controlling an important function of platelets, exocytosis. The authors demonstrate that the beta3 integrin subunit interacts with GPCR galpha13 (this was shown previously) and inhibits the downstream RhoA activation and as a result, platelet secretion.

The study is well done from the perspective of platelet biology as well as from the mechanistic standpoint. The results are unique and open a number of possibilities for future drug design. I have, however, suggestions to make the conclusions more pathologically relevant, including potential clinical importance.

1. Additional points are needed for Fig.6g due to the high variability
2. The use of additional methods to monitor platelet secretion in vivo and its consequences is highly recommended.
3. It is essential to know how beta3 inhibitors, especially ones still used in clinical practice, might affect the proposed interaction and platelet secretion.

Reviewer #2 (Remarks to the Author):

In this study, Du and coworkers demonstrate a novel cross-talk pathway in platelet activation. Their results could potentially be impactful. Experimentally, the studies seem well carried out. However, I think the paper could use improvement, especially in presentation.

1 – The introduction is too brief to describe the system or the rationale of the work and so the paper might only be appreciated to people directly in the field. I realize that this pathway is very complicated, but maybe the authors should at least include a diagram of the inside-out and outside-in pathways and the of the various molecules.

2 - More information of EE domains are needed. Do these domains occur in other proteins and would disrupting their interactions be detrimental to other signaling pathways?

3 - Does knocking down G(alpha)13 lead to excess G(beta-gamma) subunits and could Gbg contribute to any of their observations?

4 – Could the authors clarify the basis of the two waves of granular secretion?

5 – The authors mention the inclusion of serotonin in platelet granules, and Gaq contributes to scarring after infarction. These pathways be connected?

Reviewer #3 (Remarks to the Author):

This manuscript by Zhang et al described that integrin beta3 inhibits G protein signaling in platelet by direct binding to the Ga13 G protein subunit and competing with the downstream effector of Ga13, p115RhoGEF. The author first did an in vitro competition assay to show in the presence of integrin beta3 cytoplasmic tail, the binding of Ga13 to the RGS domain of p115RhoGEF is inhibited. This integrin beta3-dependent inhibition could be mimicked by using a synthetic peptide containing the EXE motif of integrin beta3 and was also observed in the platelets. Due to the inhibitory effect on Ga13-p115RhoGEF interaction, the author further observed that integrin beta3 was able to inhibit thrombin-induced RhoA activation and the subsequent platelet granule secretion. Since the platelet-secreted serotonin Aggravates Myocardial Ischemia/Reperfusion Injury (MI/RI), the authors treated the MI/RI mice with the integrin beta3-mimetic peptide and showed that the treatment reduced the plasma level of serotonin. Overall, this is an interesting study which points out the advantage of an alternative integrin-targeting strategy, in which neither ligand-binding nor talin/kindlin-binding is the target, in

the treatment of MI/RI. In my opinion, there are a few issues that need to be clarified before this manuscript can be published on Nature Communications.

1. Does integrin beta3 need to be activated to inhibit the interaction between Gα13 and p115RhoGEF? In a previous study from Du group (Gong et al, Science, 2010), it was shown that the Gα13-integrin beta3 binding was promoted by ligand-binding. The authors can clarify this by doing the same experiments of Fig. 1d-g in the presence or absence of fibrinogen.

2. How does integrin beta 3 cytoplasmic tail inhibit the binding between Gα13 and p115RhoGEF? Does it share the same binding pocket on Gα13 with p115RhoGEF? Does integrin beta3 tail interact with p115RhoGEF?

3. Does the total protein level of p115RhoGEF and RhoA change in the integrin beta3 knockout cells compared to the WT cells? Loading control like GAPDH or actin in Fig. 1d and Fig.2a should be included.

4. In Fig. 6a-f, compared to the FAAAKL peptide, mP6 (FEEERL) clearly inhibited granule secretion. In Fig. 6g, the FAAAKL peptide should also be used as control, it is a more relevant control compared to just saline solution.

REVIEWER COMMENTS

Reviewer #1 (Remarks to the Author):

This is an interesting manuscript on the mechanisms controlling an important function of platelets, exocytosis. The authors demonstrate that the beta3 integrin subunit interacts with GPCR galpha13 (this was shown previously) and inhibits the downstream RhoA activation and as a result, platelet secretion.

The study is well done from the perspective of platelet biology as well as from the mechanistic standpoint. The results are unique and open a number of possibilities for future drug design. I have, however, suggestions to make the conclusions more pathologically relevant, including potential clinical importance.

Author: We thank the reviewer for the positive feedback and insightful suggestions.

1. Additional points are needed for Fig.6g due to the high variability

Author: To address the reviewer's comment, we have performed additional experiments to increase the confidence level and revised Fig 6g with the additional data (Saline n=14, M3mP6 n=13). The significance between the saline group and M3mP6 group reaches $**p=0.0027$. By using G*power 3.1.9.7 software, t-test post-hoc analysis, when setting α probability at 0.05, the actual power between the saline and M3mP6 groups reaches 0.8603186.

2. The use of additional methods to monitor platelet secretion in vivo and its consequences is highly recommended.

Author: To address the reviewer's comment, we have performed experiments demonstrating significant increases in platelet secretion of β -TG, an important in vivo biomarker of platelet granule secretion and a pro-inflammatory chemokine, during myocardial ischemia/reperfusion *in vivo*, which was significantly inhibited by M3mP6 HPLN (Supplemental Fig 5b).

3. It is essential to know how beta3 inhibitors, especially ones still used in clinical practice, might affect the proposed interaction and platelet secretion.

Author: We thank the reviewer for this comment. To address this question, we investigated the effect of clinically used integrin antagonist Integrilin on co-immunoprecipitation between $G\alpha_{13}$ and p115-RhoGEF in thrombin-stimulated

platelets. Our data (Supplemental Fig 3) demonstrate that Integrilin, similar to RGDS peptide as shown in Fig1f,g, enhanced the binding of $G\alpha_{13}$ to p115 following thrombin stimulation. Our data also show that Integrilin (Eptifibatid) enhanced integrin-independent platelet granule secretion, although it inhibited integrin-dependent granule secretion (Fig 4d,e).

Reviewer #2 (Remarks to the Author):

In this study, Du and coworkers demonstrate a novel cross-talk pathway in platelet activation. Their results could potentially be impactful. Experimentally, the studies seem well carried out. However, I think the paper could use improvement, especially in presentation.

Author: We thank the reviewer for the constructive feedback.

1 – The introduction is too brief to describe the system or the rationale of the work and so the paper might only be appreciated to people directly in the field. I realize that this pathway is very complicated, but maybe the authors should at least include a diagram of the inside-out and outside-in pathways and the of the various molecules.

Author: We have now revised the Introduction to describe the system and rationale of the work. We also include a schematic of integrin inside-out and outside-in signaling in Supplemental figure 1.

2 - More information of EE domains are needed. Do these domains occur in other proteins and would disrupting their interactions be detrimental to other signaling pathways?

Author: Within the β cytoplasmic domain region, we observed an ExE motif in our study, with the first and third Glu residues being conserved among most β subunits except for β_4 and β_8 . We showed that the ExE motif in β_1 , β_2 , and β_3 , is responsible for $G\alpha_{13}$ binding (Shen et al. Nature 2013, MBC 2015). In a separate on-going study, we have identified an ExE(D) motif in the N terminal RGS domain of p115RhoGEF and other $G\alpha_{13}$ -binding RhoGEFs such as PDZ and LARG (confidential unpublished data), suggesting the ExE(D) motif may be a common structure for a group of $G\alpha_{13}$ -binding proteins other than the integrin cytoplasmic domain. Thus, we do not exclude the possibility that the ExE motif peptides may also affect other proteins. However, it is important to note that no toxicity has been observed in toxicology studies of the M3mP6 HLPN (MTD >80x therapeutic dose) (Pang et al Science Translational Medicine 2020).

3 - Does knocking down G(alpha)13 lead to excess G(beta-gamma) subunits and could fbg contribute to any of their observations?

Author: To address the reviewer's comment, we have performed experiments to determine the Gβ/γ levels in Gα₁₃ WT and knockout platelets. Our results suggest that Gα₁₃ knockout platelets express normal levels of Gβ and Gγ as compared with Gα₁₃ WT platelets, as shown in the attached Western blots below. On the other hand, previous studies (Moer et al, Nature Medicine 2003) indicated that Gα₁₃ knockout in platelets did not alter the expression levels of other Gα subunits. The Gβ/γ complex has been suggested to play a role in promoting platelet activation (e.g. stimulation of PI3K-dependent signaling). Considering the total deficiency of Gα₁₃ subunit, free β/γ subunits are likely increased in Gα₁₃ KO platelets. However, our results of an inhibitory role of β₃ in Gα₁₃-p115RhoGEF signaling is unlikely to be caused by the known stimulatory roles of increased free β/γ subunits. In our study, Gα₁₃ knockout had inhibitory effect on Gα₁₃-p115RhoGEF pathway and granule secretion, which is consistent with the known effect of Gα₁₃ deficiency but opposite to increased free Gβ/γ. Importantly, the inhibitory role of β₃ integrins is supported by the data that ExE motif peptide mP6 potently inhibited Gα₁₃ interaction with both β₃ and p115RhoGEF as well as its functional consequences, which is unlikely to be caused by changing free β/γ subunit levels.

Fibrinogen as an integrin ligand is likely to promote the role of integrins in regulating Gα₁₃-p115RhoGEF pathway. This is because the regulatory effect is reversed by integrin antagonist RGDS or Integrilin, which inhibits fibrinogen binding. In our experiments using washed platelets, fibrinogen is provided endogenously by the first wave granule secretion induced by GPCR agonists, and thus is present in the system.

4 – Could the authors clarify the basis of the two waves of granular secretion?

Author: Most platelet agonists (collagen, thrombin, TXA2 etc) induce early platelet granule secretion, which precedes and promotes integrin activation and platelet aggregation. This is the first wave of granule secretion, which is integrin independent. In the case of GPCR agonist thrombin and TXA2, this wave of platelet secretion is mediated by $G_{\alpha_{13}}$ (which activates the RhoA signaling pathway) and G_{α_q} (which activates PLC β and subsequent calcium mobilization/PKC activation) as well as G_{α_i} (which diminishes the inhibitory effect of cAMP and activates PI3K pathways) leading to activation of SNARE proteins and fusion of granule membrane with the plasma membranes. At relatively low concentrations of agonist (physiological condition), this wave of platelet secretion is incomplete. Most of the granules remain unreleased. GPCR agonists (including agonists such as ADP released from the granules) induce inside-out signaling and activation of integrins, which mediates platelet adhesion/aggregation and integrin outside-in signaling. Outside-in signaling induces the so-called aggregation-dependent (or integrin-dependent) granule secretion (likely mediated by the outside-in signaling-mediated activation of ITAM pathway leading to PLC γ activation, calcium mobilization and PKC activation). This is the second wave of secretion, which releases much more granule contents if the first wave secretion did not deplete all granules. In the case of some agonists, such as thrombin, the first and second secretion waves often overlap, forming a wide peak (you may also see two distinct waves of secretion in some donors). TXA2 analog U46619-stimulated secretion has two distinct waves using real time ATP secretion analysis (Li et al JBC 2003).

5 – The authors mention the inclusion of serotonin in platelet granules, and G_{α_q} contributes to scarring after infarction. These pathways be connected?

Author: Serotonin is produced by the GI tract and taken up by platelets in the bloodstream, where it is stored in dense granules. Release of serotonin from platelets during MI/R was shown to exacerbate injury to the cardiac tissue. Since serotonin is an agonist of the G_{α_q} -coupled serotonin receptor, it is likely to contribute to the role of G_{α_q} in tissue damage and scarring after infarction, although we have not directly studied this topic.

Reviewer #3 (Remarks to the Author):

This manuscript by Zhang et al described that integrin beta3 inhibits G protein signaling in platelet by direct binding to the $G_{\alpha_{13}}$ G protein subunit and competing with the downstream effector of $G_{\alpha_{13}}$, p115RhoGEF. The author first did an in vitro competition assay to show in the presence of integrin beta3 cytoplasmic tail, the binding of $G_{\alpha_{13}}$ to the RGS domain of p115RhoGEF is inhibited. This integrin beta3-dependent inhibition could be mimicked by using a synthetic peptide containing the

EXE motif of integrin beta3 and was also observed in the platelets. Due to the inhibitory effect on $G\alpha_{13}$ -p115RhoGEF interaction, the author further observed that integrin beta3 was able to inhibit thrombin-induced RhoA activation and the subsequent platelet granule secretion. Since the platelet-secreted serotonin Aggravates Myocardial Ischemia/Reperfusion Injury (MI/RI), the authors treated the MI/RI mice with the integrin beta3-mimetic peptide and showed that the treatment reduced the plasma level of serotonin. Overall, this is an interesting study which points out the advantage of an alternative integrin-targeting strategy, in which neither ligand-binding nor talin/kindlin-binding is the target, in the treatment of MI/RI. In my opinion, there are a few issues that need to be clarified before this manuscript can be published on Nature Communications.

1. Does integrin beta3 need to be activated to inhibit the interaction between $G\alpha_{13}$ and p115RhoGEF? In a previous study from Du group (Gong et al, Science, 2010), it was shown that the $G\alpha_{13}$ -integrin beta3 binding was promoted by ligand-binding. The authors can clarify this by doing the same experiments of Fig. 1d-g in the presence or absence of fibrinogen.

Author: We thank the reviewer for the positive comments. This is an excellent question. Baseline levels of integrin β_3 - $G\alpha_{13}$ interaction and its inhibitory effect on $G\alpha_{13}$ -p115RhoGEF interaction can be observed in “resting” washed platelets, but the inhibitory effect of β_3 is promoted by ligand binding to integrins. This may suggest that integrin $\alpha_{11b}\beta_3$ doesn't need to be activated to have some inhibitory effect on $G\alpha_{13}$ binding to p115RhoGEF but may also suggest that partial platelet integrin activation during in vitro platelet preparation (currently unavoidable) may have already occurred in washed “resting” platelets. However, it is clear that β_3 - $G\alpha_{13}$ interaction and its inhibitory effect on $G\alpha_{13}$ -p115RhoGEF interaction as well as its downstream signaling is greatly enhanced by activation of the ligand binding function of integrins. This is why RGDS or integrilin are similar to β_3 KO in enhancing $G\alpha_{13}$ -p115 interaction and subsequent RhoA-dependent granule secretion (Fig 1f,g and Supplemental Fig3). Fibrinogen is secreted from platelets in response to thrombin stimulation independent of integrins, and thus is present in the reaction system even without plasma or exogenous fibrinogen.

2. How does integrin beta 3 cytoplasmic tail inhibit the binding between $G\alpha_{13}$ and p115RhoGEF? Does it share the same binding pocket on $G\alpha_{13}$ with p115RhoGEF? Does integrin beta3 tail interact with p115RhoGEF?

Author: In a separate ongoing study, we are studying the detailed binding requirements of β_3 cytoplasmic domain and p115RhoGEF to $G\alpha_{13}$. We already know that the binding of integrin β_3 to $G\alpha_{13}$ requires Switch Region I of $G\alpha_{13}$.

$G\alpha_{13}$ binding to p115-RhoGEF also requires Switch Region I, which is likely the main reason for their competition. Interestingly, we also found an ExE(D) motif in p115RhoGEF and other $G\alpha_{13}$ -regulated RhoGEFs (LARG and PDZ), suggesting the possibility that ExE(D) motif is a common $G\alpha_{13}$ -binding structure. Based on protein sequence analysis (BlastP), there appears no similarity between integrin β_3 and $G\alpha_{13}$, and we have no evidence suggesting the interaction between integrin β_3 and p115RhoGEF.

3. Does the total protein level of p115RhoGEF and RhoA change in the integrin beta3 knockout cells compared to the WT cells? Loading control like GAPDH or actin in Fig. 1d and Fig.2a should be included.

Author: The total level of p115RhoGEF and RhoA do not change in WT and β_3 platelets according to the western blots of platelet lysates in Figure 1d and Figure 2a. Meanwhile, the loading control of GAPDH is now included in Supplemental Figure 2.

4. In Fig. 6a-f, compared to the FAAAKL peptide, mP6 (FEEERL) clearly inhibited granule secretion. In Fig. 6g, the FAAAKL peptide should also be used as control, it is a more relevant control compared to just saline solution.

Author: We understand the concern of the reviewer. In the previously published *Science translational Medicine* paper (Pang et al, 2020), we have performed experiments to directly compare the *in vivo* effect of saline and FAAAKL treatment, and it showed no difference between the two. Thus, we used saline as control in Fig 6g.

REVIEWERS' COMMENTS

Reviewer #1 (Remarks to the Author):

There were several relatively minor inconsistencies in the previous version of the manuscript, and these have been fixed in the revised manuscript. All my suggestions have been addressed. The manuscript seems to be well-presented and thoroughly mechanistic.

Reviewer #2 (Remarks to the Author):

I think the authors have done a good job addressing my concerns with the original manuscript.

Reviewer #3 (Remarks to the Author):

The authors have answered my questions with reasonable arguments. I have not further comments for the manuscript.